# Is the number of previous hospitalizations associated with increased in-hospital mortality after hip fracture in a developing country?

Helen Regina Mota Machareth de Morais[1], Edison Iglesias de Oliveira Vidal[2], Claudia Medina Coeli[1], Rejane Sobrino Pinheiro[1]*

1 Public Health Institute, Federal University of Rio de Janeiro, Rio de Janeiro, Brazil, 2 Botucatu Medical School, São Paulo State University, Botucatu, Brazil

* rejanesp07@gmail.com

**Data Availability Statement:** The authors do not have the permission to deposit the datasets used in this study into public domain due to ethical restrictions. Other researchers wishing to have

## Abstract

### Purpose

We aimed to examine whether the number of previous hospitalizations and the main diagnoses of those hospitalizations are associated with increased in-hospital hip fracture mortality for older people. That assessment is relevant because if those variables are shown to be associated with increased mortality, that finding could support their use as proxies for comorbidity burden for case-mix adjustment in statistical models seeking to compare the performance of hospitals regarding hip fracture mortality in settings with limited hospital information systems.

### Methods

In this retrospective cohort study of all public hospital admissions for older adults with hip fractures in the city of Rio de Janeiro between 2010 and 2011, we used data from the Hospital Admission Information System database to examine the association between in-hospital mortality and the number of hospitalizations in the previous two years and their main diagnoses through logistic regression.

### Results

Among 1938 patients included in the study there were 103 (5.3%) in-hospital deaths. Although the presence of hospitalization episodes within the two years preceding the index hip fracture was associated with increased mortality (OR: 1.78, 95%CI: 1.07 to 2.97) we did not find evidence of a gradient of increased mortality with a growing number of previous hospitalizations. Additionally, several diseases recorded as main diagnoses of previous hospitalizations were not associated with increased mortality rates, as was expected based on existing knowledge on risk factors for decreased survival in older adults with hip fractures.

access to the de-identified dataset should contact the Ethics Research Committees that approved the present study using the contact information below: Ethics Research Committee of the Public Health Institute at the Federal University of Rio de Janeiro E-mail: cep@iesc.ufrj.br Site: http://www.iesc.ufrj.br/cep/o-comite/apresentacao Telephone: +55 (21) 3938-0273 Address: Avenida Horácio Macedo, s/n Ilha do Fundão – Cidade Universitária 21941-598, Rio de Janeiro, RJ, Brazil.

**Funding:** CMC was partially supported by research fellowship grants from the National Council for Scientific and Technological Development (http://cnpq.br/) (Grant number 303295/2019-8) and Carlos Chagas Filho Foundation for Research Support in the State of Rio de Janeiro (http://www.faperj.br/) (Grant number E-26-200.003/2019). RSP was supported by research grants and research fellowship grants from the National Council for Scientific and Technological Development (http://cnpq.br/) (Grant number 481654/2012-7, 310173/2015-9and 307768/2018-0).The funders had no role in study design, data collection and analysis, decision to publish, or preparation of the manuscript.

**Competing interests:** The authors have declared that no competing interests exist.

## Conclusions

Our results suggest that, in settings where local hospital information systems have limited access to secondary diagnoses, the use of the number of previous hospitalizations or the main diagnoses associated with those hospitalizations as proxies for the profile of comorbidities of older adults with hip fractures may not be an effective way to adjust for case-mix when comparing in-hospital mortality rates among hospitals.

## Introduction

The quality of health services must be constantly evaluated and monitored to optimize their impact on population health. Health service quality encompasses aspects such as access to services, equity, technical adequacy, effectiveness, costs, safety, and patient satisfaction [1]. Performance indicators are useful when evaluating health service quality because they gauge the care process as well as its favorable or unfavorable impact on the health of individuals [1]. Performance indicators are instruments that seek to monitor the quality of health care services and work processes to identify the need for more specific evaluation when potential problems are identified [2].

In-hospital mortality after a hip fracture is an important indicator of the quality of care for a common condition that affects older adults around the world. It is considered a performance indicator and may be used to evaluate a single hospital's quality over time or to compare several hospitals against each other or a certain benchmark [2–4]. Indeed, in-hospital mortality after a hip fracture is among the 26 quality indicators that the Agency for Healthcare Research and Quality determined for the assessment of care in American hospitals [5].

Reviews of predictive factors for death following a hip fracture point to a strong relationship between death and factors such as male sex, advanced age, number of comorbidities, and the presence of cognitive deficit [6–9]. Comorbidities most closely associated with death in these studies were pulmonary diseases, heart diseases, kidney diseases, diabetes, and stroke [6–8]. Previous hospitalizations have also been considered to be associated with increased mortality following a hip fracture [10, 11].

For in-hospital mortality after hip fracture in older adults to be used as a quality indicator to compare the performance of hospitals, there are important factors that must be taken into account so that differences in findings do not primarily reflect differences in case-mix among hospitals, but rather the quality of care provided in those institutions [11–14]. Risk adjustment aims to reduce the confounding role that some variables related to case-mix, such as patients' burden of comorbidities, functional and socioeconomic status, may play on health outcomes that are used as markers of quality of care among institutions [14–18].

In Brazil, hospitalization data from the Brazilian National Health System are recorded in an administrative database, the Hospital Admission Information System, which is the most important national source of information for planning and monitoring hospital care in the country. Additionally, the Hospital Admission Information System may serve as an instrument for the assessment of the quality of inpatient care. However, up to this moment, the Brazilian Hospital Admission Information System still allows the recording of only one comorbidity and this information has been historically poorly recorded [14]. This is an important limitation of the Brazilian Hospital Admission Information System, which is shared by information systems from several other developing countries [19]. A possible strategy to attempt to overcome that important shortcoming for analyses related to the quality of

healthcare is to use data from previous hospitalizations routinely recorded in Hospital Admission Information Systems, such as the number of previous hospitalizations and the main diagnosis for each hospitalization episode, as a tool for risk adjustment in statistical models.

This study aims to analyze the association between previous hospital admissions and in-hospital mortality in older adults who underwent surgical repair of a hip fracture. It also seeks to consider the association of main diagnoses of previous hospitalizations and in-hospital death following a hip fracture. The presence of significant associations between previous hospitalizations and/or their main diagnoses and in-hospital mortality after a hip fracture could represent evidence favoring the use of those variables for risk adjustment using administrative data in scenarios typical of several developing countries, where hospital databases suffer from major shortcomings regarding the registry of comorbidities and secondary diagnoses.

## Methods

This is a retrospective cohort study of a population of elderly patients hospitalized for hip fractures in the city of Rio de Janeiro, Brazil, between January 1, 2010, and December 31, 2011. We analyzed data from the Hospital Admission Information System for all patients aged 62 years and older, whose main diagnostic codes for hospital admission according to 10th revision of the International Classification of Diseases (ICD10) were fracture of the neck of femur (S72.0), pertrochantheric fracture (S72.1), or subtrochanteric fracture (S72.2), and who underwent surgical treatment for the fracture. We excluded patients that did not undergo surgical treatment for the hip fracture and patients whose hip fracture resulted from multiple high-intensity trauma (e.g. car accident).

Causes of previous hospitalizations were identified through the main diagnosis recorded in the Hospital Admission Information System for hospitalizations that took place up to two years before the hospitalization for the index hip fracture. We recovered this information through probabilistic record linkage of the Hospital Admission Information System data from patients aged 62 years and older hospitalized for hip fractures in 2010 and 2011 and the same database for patients aged 60 years and older hospitalized between 2008 and 2011 for any cause.

We used a five-stage probabilistic record linkage technique, following a strategy recommended by Camargo Jr. and Coeli [20, 21] and using the OpenReclink software (version: 3.1) (http://reclink.sourceforge.net/). Previous research in a similar setting showed 99.4% specificity, 85.5% sensibility, 98.1% positive predictive value, and 94.9% negative predictive value for correct matching of records using this methodological approach [22].

We described categorical data as absolute numbers and proportions. Continuous data were described as mean and standard deviation (SD) when their distribution was approximately normal, or otherwise as median and interquartile ranges (IQR). We assessed distributions of continuous data for normality by inspecting their histograms.

We analyzed the association between in-hospital mortality after a hip fracture and previous hospitalizations based on three ways of classifying the latter: according to the existence of previous hospitalizations (yes or no), according to the number of previous hospitalizations (categorized as 0, 1, and 2 or more), and according to the main diagnosis of previous hospitalizations. For the last classification, we grouped the main diagnoses based on the scientific literature on risk factors for mortality following a hip fracture [6–8, 18].

We used Pearson's chi-square test to evaluate the association between the dependent variable (in-hospital mortality) and the following variables in simple analyses: presence and number of previous hospitalizations within the two years before the index fracture, sex, age (classified in age groups from 60 to 69 years, 70 to 79 years, 80 to 89 years and 90 years or

older) and type of hip fracture (classified as femoral neck fracture, pertrochantheric fracture, and subtrochanteric fracture). We calculated the odds ratio of in-hospital death associated with previous hospitalizations through two multivariable logistic regression models [23], adjusted for sex, age, and type of fracture. In one model the occurrence of previous hospitalizations over the last two years was coded as a dichotomous variable and in the other model as the total number of previous hospitalizations within that same time frame. We did not adjust those regression models for the number or type of main diagnoses from previous hospitalizations because too much collinearity would be expected with the variable encoding the number of previous hospitalizations.

We also evaluated the association between in-hospital mortality after a hip fracture and the different causes of previous hospitalizations using simple and multivariable logistic regressions. For the multivariable regression models, all groups of causes of previous hospitalizations were included and adjusted for patients' sex, age, and type of fracture.

We assessed the possibility of sparse data bias in our analyses by examining the frequency of outcome events per each category of each variable used in our models and by comparing the results of our logistic regressions with the results from penalized logistic regressions performed using the data augmentation method recommended by Greenland, Mansournia and Altman [24]. That method involved the use of a conservative F-distribution prior with a 95% odds ratio interval equivalent to 1/39 to 39, which reflects the fact that such a range encompasses most associations observed in epidemiologic studies. Whenever we found evidence of sparse data bias for any given variable, we reported the odds ratio estimates and confidence intervals from logistic regressions that penalized those variables, as described above.

We used Stata9® software to perform all analyses. We used a two-tailed alpha value of 0.05 to define statistical significance.

## Ethical approval

Access to the databases used in this study was granted by the Rio de Janeiro Municipal Health Secretary, after approval by the Ethics Review Committees from the Public Health Institute and the Municipal Health Secretary, under processes 44114515.7.0000.5286 and 15903313.0.3001.5279, respectively.

Following Brazilian regulation for ethics in research, the ethics committees waived the requirement for informed consent because obtaining informed consent would have been impossible or impracticable since this was an observational retrospective study based on secondary data from patients who were no longer under follow-up at the evaluated hospitals. The probabilistic record linkage process was carried out using identified databases because that method [20, 21] required the names of patients as an important source of information for the linkage between databases. Information was anonymized and de-identified before analysis. Only the research team had access to those databases, which were stored in a secure server at the Public Health Institute.

## Results

There were 2046 records of patients aged 62 years and older admitted to hospitals of the Brazilian Public Health System because of a hip fracture between 2010 and 2011 in the city of Rio de Janeiro. We excluded 94 (4.59%) patients because they had not been submitted to surgical repair of the fracture and 14 (0.68%) patients because their hip fracture resulted from multiple high-intensity trauma. The mean age of the remaining 1938 patients included was 79.4 years (SD: 8.3). The median length of stay of patients was 15 days (IQR: 10 to 22 days). There were 103 episodes of in-hospital death corresponding to a 5.3% mortality rate. There were no

**Table 1. Comparison of patient characteristics according to the occurrence of in-hospital mortality after a hip fracture.**

| Variable | Alive | | Death | | P* |
|---|---|---|---|---|---|
| | **N** | **%** | **N** | **%** | |
| **Sex** | | | | | 0.65 |
| Female | 1,334 | 94.5 | 77 | 5.5 | |
| Male | 501 | 95.1 | 26 | 4.9 | |
| **Age (years)** | | | | | < 0.01 |
| 60–69 | 265 | 98.1 | 5 | 1.9 | |
| 70–79 | 638 | 97.0 | 20 | 3.0 | |
| 80–89 | 742 | 94.2 | 46 | 5.8 | |
| ≥ 90 | 190 | 85.6 | 32 | 14.4 | |
| **Fracture Type** | | | | | 0.04 |
| Cervical | 1027 | 93.5 | 71 | 6.5 | |
| Pertrochantheric | 596 | 96.3 | 23 | 3.7 | |
| Subtrochanteric | 212 | 95.9 | 9 | 4.1 | |
| **Previous Hospitalization** | | | | | 0.08 |
| No | 1575 | 95.0 | 82 | 5.0 | |
| Yes | 260 | 92.5 | 21 | 7.2 | |
| **Number of Previous Hospitalizations** | | | | | 0.27 |
| 0 | 1566 | 95.0 | 82 | 5.0 | |
| 1 | 198 | 92.5 | 16 | 7.5 | |
| ≥ 2 | 71 | 93.4 | 5 | 6.6 | |
| **Total** | **1835** | **94.7** | **103** | **5.3** | |

* $\chi^2$ test.

missing data regarding the variables used in our analyses. Table 1 displays information regarding the distribution of patients regarding sex, age groups, type of fracture, and number of hospitalizations both in overall terms and concerning the occurrence of in-hospital death. Table 2 shows the frequency of the main diagnoses recorded for previous hospitalizations episodes. Table 3 displays the results of the two multivariable logistic regression models examining the association between in-hospital mortality and previous hospitalizations within the last two years. Table 4 shows the results of simple and multivariable logistic regression models examining the association between main diagnoses from previous hospitalizations within the last two years and in-hospital death after a hip fracture.

## Discussion

This is one of the largest epidemiological studies examining the in-hospital hip fracture mortality in Brazil [25–32]. Our study aimed to analyze the association between previous hospitalizations and their main diagnoses with in-hospital mortality after hip fracture surgery in older adults. Our main finding was that the occurrence of previous hospitalizations within the last two years was significantly associated with in-hospital mortality but paradoxically the number of hospitalizations in that period was not associated with a gradient of increased in-hospital mortality. On the one hand, the relatively low number of individuals who had been hospitalized two or more times in our cohort could explain this phenomenon because of a lack of statistical power. On the other hand, the low number of individuals in that category might also be pointing towards the presence of selection bias, where individuals with worse health status are denied surgical treatment or even hospitalization for hip fracture, as has been discussed by

**Table 2. Frequency of main diagnoses recorded for hospitalization episodes within the two years preceding the index hip fracture.**

| Diagnoses | Alive | | Dead | |
|---|---|---|---|---|
| | N | % | N | % |
| **Fractures** | | | | |
| Hip fracture | 21 | 100.0 | 0 | 0.0 |
| Other fractures and lesions | 74 | 91.4 | 7 | 8.6 |
| **Cardiovascular** | | | | |
| Cerebrovascular disease | 13 | 92.9 | 1 | 7.1 |
| Ischemic heart diseases | 4 | 80.0 | 1 | 20.0 |
| Peripheral Vascular Disease | 9 | 100.0 | 0 | 0.0 |
| Cardiac valve disease and congestive heart failure | 8 | 88.9 | 1 | 11.1 |
| **Neoplasms** | | | | |
| Lymphoma and leukemia | 10 | 100.0 | 0 | 0.0 |
| Malignant neoplasms | 26 | 92.9 | 2 | 7.1 |
| **Urinary System** | | | | |
| Severe and moderate kidney disease | 3 | 60.0 | 2 | 40.0 |
| Urinary tract infections | 17 | 85.0 | 3 | 15.0 |
| **Respiratory System** | | | | |
| Pneumonia | 14 | 82.4 | 3 | 17.6 |
| Tuberculosis | 3 | 100.0 | 0 | 0.0 |
| Chronic obstructive pulmonary disease | 4 | 100.0 | 0 | 0.0 |
| **Metabolic, Nutritional and Hematologic Diseases** | | | | |
| Anemias | 8 | 100.0 | 0 | 0.0 |
| Diabetes mellitus | 6 | 100.0 | 0 | 0.0 |
| Other metabolic, nutritional and hematologic diseases | 6 | 75.00 | 2 | 25.0 |
| **Other** | | | | |
| Chronic liver disease | 0 | 0.0 | 1 | 100.0 |
| Connective tissue disease | 8 | 100.0 | 0 | 0.0 |
| Osteoarthritis | 10 | 100.0 | 0 | 0.0 |
| Infectious, gastrointestinal and other diseases | 49 | 98.0 | 1 | 2.0 |
| Schizophrenia | 1 | 100.0 | 0 | 0.0 |
| Vision disorders | 16 | 100.0 | 0 | 0.0 |
| Ulcers | 2 | 100.0 | 0 | 0.0 |

others [33, 34]. The latter hypothesis is especially plausible given the fact that about 5% of the original sample were excluded because patients did not receive surgical treatment. Hence, our results suggest that the number of hospitalizations may not be a good proxy of the health status of older patients undergoing hip fracture surgery for risk adjustment in statistical models in contexts prone to the possibility of selection bias. Therefore, previous hospitalizations should be used with caution for risk adjustment of in-hospital mortality after hip fracture surgery in settings of Hospital Admission Information Systems with limited availability of data regarding comorbidities and secondary diagnoses.

To some extent, our findings disagree with those of other studies [10, 11]. For instance, Meyer et al. from Oslo, Norway [10], found that among 248 older adults hospitalized due to a hip fracture, patients who had two or more hospitalizations in the two years preceding the hip fracture were four times more likely to die within one year than patients who had not been previously hospitalized. However, that study identified a clear dose-response gradient with higher odds of mortality for those with two or more hospitalizations than for that one episode of

**Table 3. Simple and multivariable logistic regression models assessing the in-hospital mortality after hip fracture according to the occurrence of previous hospitalizations or the number of previous hospitalizations within the two years preceding the index hip fracture.**

| Variables | Crude OR (95% CI) | Adjusted OR (95% CI) | Adjusted OR (95% CI) |
|---|---|---|---|
| **Sex** | | | |
| Male | 0.90 (0.57 to 1.42) | 1.05 (0.66 to 1.68) | 1.07 (0.67 to 1.71) |
| Female | - | - | - |
| **Age (years)** | | | |
| 60–69 | - | - | - |
| 70–79 | 1.30 (0.56 to 3.00) | 1.34 (0.58 to 3.09) | 1.75 (0.65 to 9.49) |
| 80–89 | 2.56 (1.18 to 5.59) | 2.82 (1. 29 to 6.18) | 3.70 (1.44 to 9.31) |
| $\geq$ 90 | 6.84 (3.03 to 15.44) | 8.21 (3.60 to 18.77) | 10.96 (4.13 to 29.07) |
| **Fracture Type** | | | |
| Femoral neck | - | - | - |
| Pertrochanteric | 0.57 (0.36 to 0.91) | 0.47 (0.29 to 0.77) | 0.47 (0.29 to 0.77) |
| Subtrochanteric | 0.64(0.32 to 1.26) | 0.58 (0.28 to 1.18) | 0.57 (0.28 to 1.18) |
| **Previous Hospitalization*** | | | |
| Yes | 1.49 (0.91 to 2.45) | 1.78 (1.07 to 2.97) | |
| No | - | - | |
| **Number of Previous Hospitalizations*** | | | |
| 0 | - | | - |
| 1 | 1.54 (0.89 to 2.69) | | 1.69 (0.95 to 2.99) |
| $\geq$ 2 | 1.34 (0.53 to 3.42) | | 1.82 (0.70 to 4.74) |

OR: odds ratio; CI: confidence interval.

hospital admission. Aigner et al. [11] from Germany also observed that the odds of dying within six months and one year after a hip fracture was approximately two times higher in patients who had been hospitalized within the three months preceding the index hip fracture

**Table 4. Simple and multivariable logistic regression models for in-hospital mortality after hip fracture, according to the main causes of hospitalizations within the two years preceding the index hip fracture.**

| Variable | Crude OR (95% CI) | Adjusted* OR (95% CI) |
|---|---|---|
| Cerebrovascular diseases | 1.22 (0.22 to 6.70) | 1.84 (0.28 to 12.00) |
| Severe and moderate kidney disease | 6.18 (1.02 to 37.51) | 9.05 (1.29 to 63.21) |
| Cardiac valve disease and congestive heart failure | 1.64 (0.27 to 10.13) | 1.50 (0.24 to 9.48) |
| Urinary tract infections | 2.64 (0.78 to 8.97) | 3.05 (0.86 to 10.83) |
| Infectious and gastrointestinal diseases | 0.49 (0.11 to 2.16) | 0.55 (0.12 to 2.49) |
| Ischemic heart diseases | 2.39 (0.31 to 18.18) | 3.18 (0.35 to 29.00) |
| Metabolic, nutritional and hematologic diseases | 3.86 (0.77 to 19.36) | 3.27 (0.62 to 17.19) |
| Other fractures and lesions | 1.66 (0.76 to 3.64) | 1.86 (0.82 to 4.21) |
| Neoplasms | 1.28 (0.34 to 4.80) | 1.06 (0.27 to 4.12) |
| Pneumonia | 3.10 (0.78 to 16.71) | 2.8 (0.75 to 10.50) |

OR: odds ratio; CI: confidence interval.

* The Multivariable logistic regression model was adjusted for sex, age, and type of fracture.

Note: Due to a lack of variability, we were not able to estimate the OR for the following causes: anemias, arrhythmias, and conduction disorders, diabetes mellitus, chronic liver disease, chronic obstructive pulmonary disease, peripheral vascular disease, connective tissue disease, schizophrenia, hip fracture, lymphoma and leukemia, tuberculosis, ulcers, prostatic hyperplasia, and vision disorders.

when compared with patients who had not had a similar experience. However, that study did not assess for the presence of a dose-response gradient regarding mortality and their analyses did not show statistically significant differences regarding in-hospital mortality between those who had been hospitalized within three months before fracture and those who did not.

Regarding our assessment of causes of previous hospitalizations as a proxy for patients' comorbidities, our results disclosed significant associations with in-hospital mortality only for one disease group, namely severe and moderate kidney disease. Although that association is in agreement with the medical literature concerning risk factors for mortality after a hip fracture, the finding of no association between other equally relevant disease groups and mortality (e.g. dementia) is not [6–8, 18, 35–37], and signals the inherent limitations of using the main diagnoses of previous hospitalizations as a proxy and single source of information about the burden of comorbidities of patients. Such a phenomenon is likely to have occurred because the process of ascertaining the main diagnosis for an episode of hospitalization usually favors acute diagnoses over more chronic baseline comorbidities. For instance, an older adult suffering from dementia who was hospitalized because of aspiration pneumonia will usually have the latter diagnosis recorded as the main diagnosis for hospital admission instead of dementia.

Our study has several limitations. Our analyses were restricted to hospitalizations funded by the Brazilian National Health System and we were not able to access data from privately funded hospitalizations. However, recent estimates suggest that only 7.5% of older adults in Brazil have any form of private health insurance [38]. Therefore, our data is probably consistent with the health care provided to the majority of the older people in the city of Rio de Janeiro. Additionally, our limited database did not allow for a comparison between statistical models using previous hospitalizations and their main diagnoses as proxies for the burden of comorbidity of patients with models using more complete data about secondary diagnoses of patients. Finally, we were not able to analyze long term mortality outcomes beyond the hospitalization period because this would require having access to the Mortality Information System database, which was beyond the scope of this manuscript.

In conclusion, our results suggest that, in settings where local hospital information systems have limited access to secondary diagnoses, the use of the number of previous hospitalizations or the main diagnoses associated with those hospitalizations as proxies for the profile of comorbidities of older adults with hip fractures may not be an effective way to adjust for case-mix when comparing in-hospital mortality rates among hospitals.

## Supporting information

**S1 Checklist.**
(PDF)

## Acknowledgments

The authors wish to express their gratitude to Prof. Fernanda Fukushima and Dr. Raphael Vidal for their helpful comments regarding our manuscript.

## Author Contributions

**Conceptualization:** Helen Regina Mota Machareth de Morais, Edison Iglesias de Oliveira Vidal, Claudia Medina Coeli, Rejane Sobrino Pinheiro.

**Formal analysis:** Helen Regina Mota Machareth de Morais, Edison Iglesias de Oliveira Vidal, Rejane Sobrino Pinheiro.

**Funding acquisition:** Claudia Medina Coeli, Rejane Sobrino Pinheiro.

**Investigation:** Helen Regina Mota Machareth de Morais, Edison Iglesias de Oliveira Vidal, Claudia Medina Coeli, Rejane Sobrino Pinheiro.

**Methodology:** Helen Regina Mota Machareth de Morais, Edison Iglesias de Oliveira Vidal, Claudia Medina Coeli, Rejane Sobrino Pinheiro.

**Supervision:** Rejane Sobrino Pinheiro.

**Validation:** Helen Regina Mota Machareth de Morais, Edison Iglesias de Oliveira Vidal, Rejane Sobrino Pinheiro.

**Visualization:** Helen Regina Mota Machareth de Morais.

**Writing – original draft:** Helen Regina Mota Machareth de Morais.

**Writing – review & editing:** Helen Regina Mota Machareth de Morais, Edison Iglesias de Oliveira Vidal, Claudia Medina Coeli, Rejane Sobrino Pinheiro.

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
