## [Editor Report · Decision Letter 0]

5 Sep 2019

PONE-D-19-23508

Is the number of previous hospitalizations associated with increased in-hospital mortality after hip fracture in a developing country?

PLOS ONE

Dear Mrs de Morais,

Thank you for submitting your manuscript to PLOS ONE. After careful consideration, we feel that it has merit but does not fully meet PLOS ONE’s publication criteria as it currently stands. Therefore, we invite you to submit a revised version of the manuscript that addresses the points raised during the review process.

We would appreciate receiving your revised manuscript by Oct 20 2019 11:59PM. To enhance the reproducibility of your results, we recommend that if applicable you deposit your laboratory protocols in protocols.io, where a protocol can be assigned its own identifier (DOI) such that it can be cited independently in the future. For instructions see: http://journals.plos.org/plosone/s/submission-guidelines#loc-laboratory-protocols

We look forward to receiving your revised manuscript.

Kind regards,

Felipe Hada Sanders, M.D.

Academic Editor

PLOS ONE

Journal Requirements:

2. In ethics statement in the manuscript and in the online submission form, please provide additional information about the patient records used in your retrospective study. Specifically, please ensure that you have discussed whether all data were fully anonymized before you accessed them and/or whether the IRB or ethics committee waived the requirement for informed consent. If patients provided informed written consent to have data from their medical records used in research, please include this information.

Additional Editor Comments (if provided):

beautifully written. please attach the ethics comittee approval.
---

## [Author Response · Author response to Decision Letter 0]

7 Oct 2019

Journal Requirements

1. Please ensure that your manuscript meets PLOS ONE's style requirements, including those for file naming. The PLOS ONE style templates can be found at http://www.journals.plos.org/plosone/s/file?id=wjVg/ PLOSOne_formatting_sample_main_body.pdf and http://www.journals.plos.org/plosone/s/file?id=ba62/ PLOSOne_formatting_sample_title_authors_affiliations.pdf

Authors' response:

We revised the manuscript to comply with PLOS ONE's style requirements including those for file naming.

2. In ethics statement in the manuscript and in the online submission form, please provide additional information about the patient records used in your retrospective study. Specifically, please ensure that you have discussed whether all data were fully anonymized before you accessed them and/or whether the IRB or ethics committee waived the requirement for informed consent. If patients provided informed written consent to have data from their medical records used in research, please include this information.

Authors' response:

In the revised version of our manuscript we have provide more detailed information regarding the patients’ records used in our study. The new section of our manuscript dedicated to the ethical approval of our investigation reads as follows:

“The Ethics Review Committees of the Public Health Institute and the Municipal Health Secretary approved this study under the processes 44114515.7.0000.5286 and 15903313.0.3001.5279, respectively. Following Brazilian regulation for ethics in research the ethics committees waived the requirement for informed consent because obtaining informed consent would have been impossible or impracticable since this was an observational retrospective study based on secondary data from patients who were no longer under follow-up at the evaluated hospital units. The probabilistic record linkage process was carried out using identified databases because that method [20-21] required the names of patients as an important source of information for the linkage between databases. Information was anonymized and de-identified prior to analysis. Only the research team had access to those databases, which were stored in a secure server at the Public Health Institute.”

Additional Editor Comments

beautifully written. please attach the ethics comittee approval.

Authors' response:

Thank you for your generous comments. We have uploaded the reports with the approval of our study by both Ethics Review Committees to the Editorial Manager Submission System. 

Reviewer’s Comments

P.9, line 110 of the attached file: “Please attach this approval”

Authors' response:

We have uploaded the reports with the approval of our study by both Ethics Review Committees to the Editorial Manager Submission System. Please note that we have also corrected the identification numbers of both processes at each Ethics Review Committees.

---

## [Decision Letter · Decision Letter 1]

17 Jan 2020

PONE-D-19-23508R1

Is the number of previous hospitalizations associated with increased in-hospital mortality after hip fracture in a developing country?

PLOS ONE

Dear Mrs de Morais,

Thank you for submitting your manuscript to PLOS ONE. After careful consideration, we feel that it has merit but does not fully meet PLOS ONE’s publication criteria as it currently stands. Therefore, we invite you to submit a revised version of the manuscript that addresses the points raised during the review process.

The authors are required to respond to the reviewers comments and make all necessary changes.

We would appreciate receiving your revised manuscript by Mar 02 2020 11:59PM. To enhance the reproducibility of your results, we recommend that if applicable you deposit your laboratory protocols in protocols.io, where a protocol can be assigned its own identifier (DOI) such that it can be cited independently in the future. For instructions see: http://journals.plos.org/plosone/s/submission-guidelines#loc-laboratory-protocols

We look forward to receiving your revised manuscript.

Kind regards,

Osama Farouk

Academic Editor

PLOS ONE

Reviewers' comments:

Reviewer's Responses to Questions

**Comments to the Author**

1. If the authors have adequately addressed your comments raised in a previous round of review and you feel that this manuscript is now acceptable for publication, you may indicate that here to bypass the “Comments to the Author” section, enter your conflict of interest statement in the “Confidential to Editor” section, and submit your "Accept" recommendation.

Reviewer #1: (No Response)

Reviewer #2: (No Response)

2. Is the manuscript technically sound, and do the data support the conclusions?

Reviewer #1: (No Response)

Reviewer #2: Yes

3. Has the statistical analysis been performed appropriately and rigorously? 

Reviewer #1: (No Response)

Reviewer #2: I Don't Know

4. Have the authors made all data underlying the findings in their manuscript fully available?

Reviewer #1: (No Response)

Reviewer #2: Yes

5. Is the manuscript presented in an intelligible fashion and written in standard English?

Reviewer #1: (No Response)

Reviewer #2: No

6. Review Comments to the Author

Reviewer #1: General comments:

It’s interesting to study the leading factors of mortality among hip fracture patients in addition to medical and public health importance, but this study idea titled “Is the number of previous hospitalizations associated with increased in-hospital mortality after hip fracture in a developing country?” raises some concerns.

Unfortunately, the hypothesis and research question about the relation between one main diagnosis "what was found in records" and mortality in those patients is not based in scientific basis that we couldn't find in the introduction or discussion sections. Hip fracture is a condition mainly related to elderly, the mean age of patients in this study is 79.4 years (SD: 8.3) with expected complicated medical conditions in the past two years from the index hip fracture. With no doubt before conduction of this study, one reported diagnosis of previous hospitalizations is not the only related factor to the mortality among those patients with more than one morbidity condition in this study

In addition, there are major defects in the manuscript and not well written especially in methods, data analysis and results sections. However, I have provided some remarks below.

Abstract:

In conclusion section: the word correlation is not correct to be written all over the manuscript, better to say associated factors or correlates.

The sentences in lines 20 to 24 are not clear and not focused on the study aims, as “ evaluate , performance of hospitals, limited information system.

“Correlates”: is mentioned in the title and didn’t mentioned after words anywhere all over the manuscript.

In methods section: no data were mentioned about data collection

The logistic regression analysis was not mentioned and its results “ which is not clear in analysis section after that”

Introduction

• Title and aim of the study are not matched with the introduction

• First two paragraphs are not related to the title nor the aim of the study

• Page 4, line 62: this study was not done to evaluate the quality of health care in hip fracture patients.

• No need to write about evaluation of health care in hp fracture patients, this could be mentioned in one sentence.

• Page 5, line 72 to 75: “Additionally, the national Hospital Admission Information System may serve as an instrument for the assessment of the quality of inpatient care. However, up to this moment the Brazilian Hospital Admission Information System still allows the recording of only one comorbidity and this information has been historically recorded poorly”….. this sentence about the poor data source is talking about a deficient tool to do this study,

Methods:

• Non concurrent: corrected to be retrospective cohort study

• Between 2010 and 2011: could be corrected to be “ from the start of 2010 to the end of 2011” or whatever the included months.

• Mention the level of significance of p value

• Please mention the details of sensitivity analysis that was referred to in results section.

Results:

- Median of hospital stay duration: is this variable is non parametric?? Please clarify!!

- Presentation of results is not well written regarding tables 2,3 and 4, the titles only were mentioned

- Page 9, line 66 sensitivity analysis is firstly to be mentioned here. This analysis was not mentioned in methods section and no details here were presented.

- N in tables: better to be corrected to “no.”

- Table (1):

• Total column could be moved after p value column for better understanding.

• No. of previous hospitalization: is better to be grouped to 3 groups: “0, 1 and 2 and more”. Qui square test could be done correctly without “0” in any cell.

• Write the test of significance as a footnote under the table.

- Table (2): add number of mortalities in each diagnosis.

- Table 3 and 4:

- In general, the number of observations is lower than needed to carry logistic regression as in case of number of previous hospitalizations and ischemic heart disease. Number of mortalities in each diagnosis is not clear while number of cases was only mentioned. In general, regression analysis models were not done on statistical basis.

• In the title “Multiple logistic regression models were adjusted for sex, age and type of fracture” please write this sentence as a footnote

Discussion:

• No discussion of the mortality incidence was found with other studies

• Discussion should be rewritten after corrections in results section to reevaluate the significant relations

Reviewer #2: Very interesting article. The major problem is the English language; abstract, introduction and methods are not clear. Seems like they have been written from a different person than the other parts.

Abstract: I would change the phrase "That assessment.....systems" is too long, and in "restrospective...2011" there is no verb

Introduction: You have analized too much what performance indicators are (I suggest to cancel 42-44 and 57-60 for example)

Methods: ethical approval is repeated: 106-107, 132-134

7. PLOS authors have the option to publish the peer review history of their article (what does this mean?). If published, this will include your full peer review and any attached files.

Reviewer #1: Yes: Dalia G Mahran

Reviewer #2: No

---

## [Author Response · Author response to Decision Letter 1]

20 Aug 2020

Reviewer’s Comments

1. If the authors have adequately addressed your comments raised in a previous round of review and you feel that this manuscript is now acceptable for publication, you may indicate that here to bypass the “Comments to the Author” section, enter your conflict of interest statement in the “Confidential to Editor” section, and submit your "Accept" recommendation.

Reviewer #1: (No Response)

Reviewer #2: (No Response)

2. Is the manuscript technically sound, and do the data support the conclusions?

Reviewer #1: (No Response)

Reviewer #2: Yes

Authors' response: We are grateful for the reviewer #2 assessment. 

3. Has the statistical analysis been performed appropriately and rigorously? 

Reviewer #1: (No Response)

Reviewer #2: I Don't Know

Authors' response: We revised the statistical methods of our manuscript and corrected our previous analyses to avoid the occurrence of sparse data bias, as described in detail in our answers to the comments of reviewer #1.

4. Have the authors made all data underlying the findings in their manuscript fully available?

Reviewer #1: (No Response)

Reviewer #2: Yes

Authors' response: Please note that in the previous version of our manuscript we had added a data availability statement to the end of the manuscript explaining that the authors do not have the permission to deposit the datasets used in this study into the public domain because even the de-identified dataset derived from the probabilistic record linkage process has the potential to be re-identified through the combination of several databases. If other researchers wish to have access to the de-identified dataset, they should contact the authors who will file an authorization request to the Ethics Review Committees that authorized this study. Once that permission is granted by both Ethics Research Committees, the authors shall be able to share the study dataset with requesting researchers. We understand that doing so is consistent with PLOS ONE policy on data availability, which recognizes that "in some instances, authors may not be able to make their underlying data set publicly available for legal or ethical reasons”.

5. Is the manuscript presented in an intelligible fashion and written in standard English?

Reviewer #1: (No Response)

Reviewer #2: No

Authors' response: An experienced English teacher revised our manuscript. Additionally, we asked for other colleagues to read our manuscript and to confirm whether it was sufficiently clear.

6. Review Comments to the Author

Reviewer #1: General comments:

It’s interesting to study the leading factors of mortality among hip fracture patients in addition to medical and public health importance, but this study idea titled “Is the number of previous hospitalizations associated with increased in-hospital mortality after hip fracture in a developing country?” raises some concerns.

Unfortunately, the hypothesis and research question about the relation between one main diagnosis "what was found in records" and mortality in those patients is not based in scientific basis that we couldn't find in the introduction or discussion sections. Hip fracture is a condition mainly related to elderly, the mean age of patients in this study is 79.4 years (SD: 8.3) with expected complicated medical conditions in the past two years from the index hip fracture. With no doubt before conduction of this study, one reported diagnosis of previous hospitalizations is not the only related factor to the mortality among those patients with more than one morbidity condition in this study

In addition, there are major defects in the manuscript and not well written especially in methods, data analysis and results sections. However, I have provided some remarks below.

Authors' response: We have read all the comments of reviewer #1 with great attention and we are grateful for the time they dedicated to evaluating our manuscript. We revised our manuscript carefully and provide below point-by-point responses to their comments.

The reviewer stated that “the hypothesis and research question about the relation between one main diagnosis "what was found in records" and mortality in those patients is not based in scientific basis that we couldn't find in the introduction or discussion sections." Respectfully, we disagree with the reviewer's remark and could not find in our manuscript any evidence corroborating their argument. Our introduction section was divided into 6 paragraphs that outlined the scientific rationale underlying our study. In the first paragraph, we explained why health service quality is an important field of research in public health and why performance indicators play an important role in the assessment of health service quality. In the second paragraph, we argued that in-hospital mortality after a hip fracture in older adults is an important indicator of the performance of care and that it is listed among the 26 quality indicators that the Agency for Healthcare Research and Quality determined for the assessment of the quality of care in American hospitals. In the third paragraph, we explored briefly some known risk factors for mortality after a hip fracture and provided references to previous studies that had found an association between previous hospitalizations and hip fracture mortality. In the fourth paragraph, we explained that "for in-hospital mortality after hip fracture in older adults to be used as a quality indicator to compare the performance of hospitals there are important factors that must be taken into account so that differences in findings do not primarily reflect differences in case-mix among hospitals, but rather the quality of care provided in those institutions". In the fifth paragraph, we introduced the reader to the Brazilian Hospital Admission Information System and to its limitation regarding the possibility to record only one comorbidity for each hospitalization episode, which is shared by information systems from other developing countries. In that paragraph, we also outlined the reasoning behind a strategy to overcome that limitation by using previous hospitalizations and the main diagnosis of each hospitalization episode as tools for risk adjustment in statistical models aiming to assess the quality of hospital health services for older adults with a hip fracture. Finally, in the sixth paragraph, we explained that in this study we aimed to evaluate the association between in-hospital mortality in older people who underwent surgical repair of a hip fracture and previous hospital admissions and the main diagnosis recorded for each hospitalization because if such associations were found they could be construed as evidence favoring the use of those variables for risk adjustment in health care quality comparisons among hospitals in developing countries whose administrative databases share similar limitations as the Brazilian Hospital Admission Information System. Hence, we believe that our introduction section thoroughly outlined the scientific basis and relevance of our research question. 

We also carefully reexamined our discussion section and found that it was consistent with the arguments laid out in our introduction section as can be seen in the following excerpts of our manuscript: 

“Hence, our results suggest that the number of hospitalizations may not be a good proxy of the health status of older patients undergoing hip fracture surgery for risk adjustment in statistical models in contexts prone to the possibility of selection bias. Therefore, previous hospitalizations should be used with caution for risk adjustment of in-hospital mortality after hip fracture surgery in settings of Hospital Admission Information Systems with limited availability of data regarding comorbidities and secondary diagnoses.” (first paragraph of the discussion section) 

“In conclusion, our results suggest that, in settings where local hospital information systems have limited access to secondary diagnoses, the use of the number of previous hospitalizations or the main diagnoses associated with those hospitalizations as proxies for the profile of comorbidities of older adults with hip fractures may not be an effective way to adjust for case-mix when comparing in-hospital mortality rates among hospitals.” (last paragraph of the discussion section)

Still, regarding the reviewer’s first comment we feel concerned about the misuse of supposed quoted citations from our text. We thoroughly searched our text and could not find the reviewer’s quote “what was found in records".

The reviewer also stated that “With no doubt before conduction of this study, one reported diagnosis of previous hospitalizations is not the only related factor to the mortality among those patients with more than one morbidity condition in this study”. Respectfully, we never claimed that previous hospitalizations were the only risk factors for hip fracture mortality in any context. As can be seen in the third paragraph of our introduction section, we are aware of the most common risk factors for hip fracture mortality. Our focus on the associations between in-hospital mortality after a hip fracture and previous hospital admissions and the main diagnoses associated with those hospitalization episodes is explained by our intent to assess whether those variables could be useful risk adjustment tools in the context of health service quality research. Unfortunately, we are deeply concerned that the reviewer's comments above might indicate that they did not understand the scientific context and rationale of our investigation.

Reviewer #1:

Abstract:

In conclusion section: the word correlation is not correct to be written all over the manuscript, better to say associated factors or correlates.

Authors' response: Respectfully, we feel puzzled and deeply concerned about the reviewer’s comment lacking any plausible relationship with the previous version of our manuscript that we had submitted to PLOS ONE. The word “correlation” was not used anywhere in our abstract or in our main text. 

Reviewer #1:

The sentences in lines 20 to 24 are not clear and not focused on the study aims, as “ evaluate , performance of hospitals, limited information system.

Authors' response: Respectfully, we disagree with the reviewer’s assessment. Please find below the whole “Purpose” section of the previous version of our abstract. 

“Purpose: We aimed to examine whether the number of previous hospitalizations and the main diagnoses of those hospitalizations are associated with increased in-hospital hip fracture mortality for older people. That assessment is relevant because if those variables are shown to be associated with increased mortality, that finding could support their use as proxies for comorbidity burden for case mix adjustment in statistical models seeking to compare the performance of hospitals regarding hip fracture mortality in settings with limited hospital information systems.”

The reviewer argued that the last sentence of that section was not clear and was unrelated to the study’s aim. Our whole manuscript was revised by an experienced English teacher, who assured us that that sentence was already clear in the last version of our manuscript. The only correction that was added to that fragment involved substituting "case-mix" for "case mix". Additionally, we believe that that sentence is essential to convey the rationale underpinning the aim of our research question.

Reviewer #1: "Correlates”: is mentioned in the title and didn’t mentioned after words anywhere all over the manuscript.

Authors' response: Respectfully, the title of our manuscript never included the word “correlates”. Our title was and still is “Is the number of previous hospitalizations associated with increased in-hospital mortality after hip fracture in a developing country?” Once again, we cannot fathom how the reviewer was able to make such a statement with no relationship with the content of our submission.

Reviewer #1: In methods section: no data were mentioned about data collection

Authors' response: We understand the reviewer's comment and recognize that the construction of clear, informative, and succinct abstracts is a balancing act, where researchers must weight which pieces of information are essential to convey the main message of their study, and which are not. In an attempt to comply with the reviewer's request for more information regarding data extraction and with PLOS ONE's 300-word limit for abstracts, we added information about the database that we used in this study to the methods section of our abstract but we were not able to include details about the variables that were extracted from that database. The new methods section of our abstract reads as follows:

“Methods: In this retrospective cohort study of all public hospital admissions for older adults with hip fractures in the city of Rio de Janeiro between 2010 and 2011, we used data from the Hospital Admission Information System database to examine the association between in-hospital mortality and the number of hospitalizations in the previous two years and their main diagnoses through logistic regression.”

Reviewer #1:

The logistic regression analysis was not mentioned and its results “ which is not clear in analysis section after that”

Authors' response: Respectfully, we disagree with the reviewer's comment. We reported in the results section of our abstract results from our logistic regression analyses, which were represented by the odds ratio (OR: 1.78, 95%CI: 1.07 to 2.97). Because of the lack of space, we were not able to present other quantitative results from our logistic regression analyses, but we believe that our main results were communicated effectively to readers. The revised results section of our abstract reads as follows.

“Results: Among 1938 patients included in the study there were 103 (5.3%) in-hospital deaths. Although the presence of hospitalization episodes within the two years preceding the index hip fracture was associated with increased mortality (OR: 1.78, 95%CI: 1.07 to 2.97) we did not find evidence of a gradient of increased mortality with a growing number of previous hospitalizations. Additionally, several diseases recorded as main diagnoses of previous hospitalizations were not associated with increased mortality rates, as was expected based on existing knowledge on risk factors for decreased survival in older adults with hip fractures.”

Reviewer #1: Introduction

• Title and aim of the study are not matched with the introduction

Authors' response: Respectfully, we disagree with the reviewer’s comment. The title of our article is “Is the number of previous hospitalizations associated with increased in-hospital mortality after hip fracture in a developing country?”. The aim of our article in our introduction section was described as “This study aims to analyze the association between previous hospital admissions and in-hospital mortality in older adults who underwent surgical repair of a hip fracture.” (Lines 78-79). Hence, we argue that our title and aim are perfectly in line with each other. 

Reviewer #1:

• First two paragraphs are not related to the title nor the aim of the study

Authors' response: Respectfully, we do not understand the basis for the reviewer’s expectation that there should be a perfect relationship between the first two paragraphs of our introduction section and the title of our manuscript. We have examined the STROBE statement and its checklist and could not find any recommendation that the aim of the study or its title should be outlined in the first two paragraphs of the introduction section. On the other hand, we firmly believe that our introduction section did comply with the STROBE Statement requirement that the background and rationale of observational studies be explained in the introduction section of manuscripts, as described in detail in our answer to the reviewer's first comment above.

Reviewer #1: 

• Page 4, line 62: this study was not done to evaluate the quality of health care in hip fracture patients.

Authors' response: We agree with the reviewer's comment

. Indeed, we never stated that our study aimed to assess the quality of healthcare for patients with hip fractures. The paragraph that began in line 62 of the previous version (with track changes) of our manuscript stated the following:

"For in-hospital mortality after hip fracture in older adults to be used as a quality indicator to compare the performance of hospitals, there are important factors that must be taken into account so that differences in findings do not primarily reflect differences in case-mix among hospitals, but rather the quality of care provided in those institutions [11–14]. Risk adjustment aims to reduce the confounding role that some variables related to case-mix, such as patients' burden of comorbidities, functional and socioeconomic status, may play on health outcomes that are used as markers of quality of care among institutions [14–18]" 

That paragraph was never meant to describe the aim of our study but simply to describe the rationale behind our aim.

Reviewer #1:

• No need to write about evaluation of health care in hp fracture patients, this could be mentioned in one sentence.

Authors' response: Respectfully, we disagree with the reviewer's comment. As described in our introduction section, the explanations about the quality of hospital health services, and performance indicators in the context of hip fractures in older adults represent essential information without which the background and rationale of our study cannot be conveyed appropriately to the reader.

Reviewer #1: • Page 5, line 72 to 75: "Additionally, the national Hospital Admission Information System may serve as an instrument for the assessment of the quality of inpatient care. However, up to this moment the Brazilian Hospital Admission Information System still allows the recording of only one comorbidity and this information has been historically recorded poorly”….. this sentence about the poor data source is talking about a deficient tool to do this study,

Authors' response: Respectfully, we disagree with the reviewer’s comment. Once again, we have the impression that the reviewer did not understand the basic aspects of the context and rationale of our study. It suffices to read the whole paragraph from which the reviewer extracted the text fragment above and its following paragraph, to understand that the rationale of our study took into account the limitations of the Brazilian Hospital Admission Information System and sought to assess whether, in the context of those limitations, previous hospitalizations were associated with in-hospital mortality. We copy the content of those two paragraphs below.

“In Brazil, hospitalization data from the Brazilian National Health System are recorded in an administrative database, the Hospital Admission Information System, which is the most important national source of information for planning and monitoring hospital care in the country. Additionally, the Hospital Admission Information System may serve as an instrument for the assessment of the quality of inpatient care. However, up to this moment, the Brazilian Hospital Admission Information System still allows the recording of only one comorbidity and this information has been historically poorly recorded [14]. This is an important limitation of the Brazilian Hospital Admission Information System, which is shared by information systems from several other developing countries [19]. A possible strategy to attempt to overcome that important shortcoming for analyses related to the quality of healthcare is to use data from previous hospitalizations routinely recorded in Hospital Admission Information Systems, such as the number of previous hospitalizations and the main diagnosis for each hospitalization episode, as a tool for risk adjustment in statistical models. 

 This study aims to analyze the association between previous hospital admissions and in-hospital mortality in older adults who underwent surgical repair of a hip fracture. It also seeks to consider the association of main diagnoses of previous hospitalizations and in-hospital death following a hip fracture. The presence of significant associations between previous hospitalizations and/or their main diagnoses and in-hospital mortality after a hip fracture could represent evidence favoring the use of those variables for risk adjustment using administrative data in scenarios typical of several developing countries, where hospital databases suffer from major shortcomings regarding the registry of comorbidities and secondary diagnoses.” 

Reviewer #1:

Methods:

• Non concurrent: corrected to be retrospective cohort study

Authors' response: We would like to point out that the term "non-concurrent cohort study" is an accepted nomenclature for the classification of our study as explained in the following text on the taxonomy of study designs: https://www.ncbi.nlm.nih.gov/books/NBK154468/ . 

Nevertheless, we followed the reviewer's recommendation and changed our text using the more commonly used wording: "retrospective cohort study".

Reviewer #1:

• Between 2010 and 2011: could be corrected to be “ from the start of 2010 to the end of 2011” or whatever the included months.

Authors' response: We followed the reviewer’s recommendation and modified our text, which now reads as follows:

“This is a retrospective cohort study of a population of elderly patients hospitalized for hip fractures in the city of Rio de Janeiro, Brazil, between January 1, 2010, and December 31, 2011.”

Reviewer #1:

• Mention the level of significance of p value

Authors' response: In compliance with the reviewer's request, we added the following text to our methods section: 

“We used a two-tailed alpha value of 0.05 to define statistical significance.” (last paragraph of the Methods section) 

Reviewer #1: 

• Please mention the details of sensitivity analysis that was referred to in results section.

Authors' response: We followed one of the reviewer's recommendations made during their comments regarding our results section below and recategorized the number of previous hospitalizations variable in table 1 into 0, 1, and 2 or more hospitalization episodes, as requested. Aiming to attain consistency in the presentation of our results, we also used that categorization scheme for the regression analyses involving that variable. Hence, there was no need to perform our previous sensitivity analysis anymore because that analysis involved exactly that recategorization.

Reviewer #1: Results:

- Median of hospital stay duration: is this variable is non parametric?? Please clarify!!

Authors' response: We have added the following text to our methods section:

“We described categorical data as absolute numbers and proportions. Continuous data were described as mean and standard deviation (SD) when their distribution was approximately normal, or otherwise as median and interquartile ranges (IQR) [20]. We assessed distributions of continuous data for normality by inspecting their histograms.” 

Thereby, we believe that the reader will understand that when we described a continuous variable using median and IQR, that variable did not follow the normal distribution.

Reviewer #1:

- Presentation of results is not well written regarding tables 2,3 and 4, the titles only were mentioned

Authors' response: We have made changes to the titles of tables 2, 3, and 4. The new titles are as follows:

Table 2. Frequency of main diagnoses recorded for hospitalization episodes within the two years preceding the index hip fracture.

Table 3. Simple and multivariable logistic regression models assessing the in-hospital mortality after hip fracture according to the occurrence of previous hospitalizations or the number of previous hospitalizations within the two years preceding the index hip fracture.

Table 4. Simple and multivariable logistic regression models for in-hospital mortality after hip fracture, according to the main causes of hospitalizations within the two years preceding the index hip fracture. 

Please, note that we reframed table 3 to make it clear that we conducted two separate multivariable regressions, one in which previous hospitalizations were treated as a dichotomous variable and another where the number of previous hospitalizations was categorized as 0, 1, and 2 or more.

Reviewer #1: 

- Page 9, line 66 sensitivity analysis is firstly to be mentioned here. This analysis was not mentioned in methods section and no details here were presented.

Authors' response: We addressed the issue of the sensitivity analysis in a previous answer. 

Reviewer #1:

- N in tables: better to be corrected to "no."

Authors' response: It is common practice in scientific journals to report the number of individuals using the letter “N” and we could not find any recommendation in PLOS ONE’s guidelines for authors to avoid that standard. Nevertheless, if PLOS ONE’s editors believe we should follow another standard, we will be happy to comply.

Reviewer #1: 

- Table (1):

• Total column could be moved after p value column for better understanding.

Authors' response: We have substituted the column with the totals by a column with the number of alive individuals. We believe that this change made it easier for readers to compare the different frequencies of in-hospital death according to the categories of each variable.

Reviewer #1: 

• No. of previous hospitalization: is better to be grouped to 3 groups: "0, 1 and 2 and more". Qui square test could be done correctly without "0" in any cell.

Authors' response: We followed the reviewer's recommendation and recategorized that variable into 0, 1, and 2 or more hospitalization episodes, as described in a previous answer.

Reviewer #1: • Write the test of significance as a footnote under the table.

Authors' response: The Chi-square test of significance had already been added as a footnote to table 1 in the previous version of our manuscript.

Reviewer #1:

- Table (2): add number of mortalities in each diagnosis.

Authors' response: We followed the reviewer’s request and formatted table 2 according to the same standard used in table 1.

Reviewer #1:

- Table 3 and 4:

- In general, the number of observations is lower than needed to carry logistic regression as in case of number of previous hospitalizations and ischemic heart disease. Number of mortalities in each diagnosis is not clear while number of cases was only mentioned. In general, regression analysis models were not done on statistical basis.

Authors' response: We are very grateful for the reviewer's comment that allowed us to recognize that our previous analyses had incurred in the problem of sparse data bias (see Greenland et al 2016 https://www.bmj.com/content/352/bmj.i1981). We followed Greenland's et al recommendation and assessed the presence of sparse data bias in our results by examining the frequencies of outcome events per variable and by comparing the results of our logistic regressions with the results from penalized logistic regressions performed using the data augmentation method proposed by those authors using an F-distribution prior with a 95% odds ratio interval equivalent to 1/39 to 39. Whenever we found evidence of sparse data bias for any given variable, we reported the odds ratio estimates and confidence intervals from logistic regressions that penalized those variables using the approach recommended by Greenland et al 2016. That approach was described in the methods section of the new version of our manuscript.

• In the title “Multiple logistic regression models were adjusted for sex, age and type of fracture” please write this sentence as a footnote

Authors' response: In the previous clean version (i.e. without track changes) of our manuscript, we had already reported that information as a footnote for tables 3 and 4. However, we apologize for our mistake that this change was not apparent in the previous version of our manuscript with track changes, which must be the one that the reviewer assessed.

Reviewer #1:

Discussion:

• No discussion of the mortality incidence was found with other studies

Authors' response: The in-hospital mortality rate of our study was 5.3%, whereas the in-hospital mortality reported by other Brazilian studies of hip fracture in older adults ranged between 3.8% and 14.6% (https://www.ncbi.nlm.nih.gov/pmc/articles/PMC3971362/ ; https://pubmed.ncbi.nlm.nih.gov/16871434/ ; https://www.scielosp.org/article/rsp/2015.v49/12/ ). The in-hospital mortality in our study was similar to that reported in southern Ontario, Canada (5.0%) (https://www.ncbi.nlm.nih.gov/pmc/articles/PMC2947119/) and higher than that reported for intertrochanteric fractures in the US (1.7%) (https://pubmed.ncbi.nlm.nih.gov/28255840/). However, we felt that it would not be appropriate to deviate the focus of our discussion beyond the aims of our study. We believe that if we deviated our discussion from the main objectives of our study to address the possible reasons behind the different mortality rates among epidemiological studies of hip fracture in Brazil and elsewhere, our discussion section would become much lengthier but would not be substantially improved to justify such a decision. Furthermore, we recognize that although PLOS ONE does not impose any length limits on the articles that it publishes, it does recommend that study findings be presented and discussed concisely (https://journals.plos.org/plosone/s/submission-guidelines). Furthermore, the STROBE statement recommends that the discussion section summarizes "key results with reference to study objectives", which we did.

Reviewer #1: • Discussion should be rewritten after corrections in results section to reevaluate the significant relations

Authors' response: The changes that our results underwent after the consideration of the presence of sparse data bias and the implementation of the penalized logistic regression methods were incorporated into our discussion section bud did not change the overall interpretation of our findings and did not require major changes to our discussion.

Reviewer #2: Very interesting article. 

Authors' response: We are grateful for the reviewer’s generous comment.

Reviewer #2:

The major problem is the English language; abstract, introduction and methods are not clear. Seems like they have been written from a different person than the other parts.

Authors' response: An experienced English teacher revised our manuscript. Additionally, we asked for other colleagues to read our manuscript and to confirm whether it was sufficiently clear.

 Reviewer #2: 

Abstract: I would change the phrase "That assessment.....systems" is too long, and in "restrospective...2011" there is no verb

Authors' response: We understand that writing an informative and yet concise abstract is always challenging. We agree that that is a long sentence. However, we have confirmed that it is sufficiently clear and we firmly believe that it conveys an essential piece of information regarding the rationale underpinning our study.

Reviewer #2: 

Introduction: You have analized too much what performance indicators are (I suggest to cancel 42-44 and 57-60 for example)

Authors' response: We firmly believe that those sentences are essential to convey important background information about the relevance and the rationale of our study. Deleting those sentences would jeopardize the understanding of our investigation aims. 

Reviewer #2: Methods: ethical approval is repeated: 106-107, 132-134

Authors' response: We followed the reviewer's recommendation and restricted the information about the ethics review committee's approval to a single section of our manuscript.

---

## [Decision Letter · Decision Letter 2]

23 Sep 2020

Is the number of previous hospitalizations associated with increased in-hospital mortality after hip fracture in a developing country?

PONE-D-19-23508R2

Dear Dr. Pinheiro,

We’re pleased to inform you that your manuscript has been judged scientifically suitable for publication and will be formally accepted for publication once it meets all outstanding technical requirements.

Kind regards,

Osama Farouk

Academic Editor

PLOS ONE

Additional Editor Comments (optional):

Reviewers' comments:

Reviewer's Responses to Questions

**Comments to the Author**

1. If the authors have adequately addressed your comments raised in a previous round of review and you feel that this manuscript is now acceptable for publication, you may indicate that here to bypass the “Comments to the Author” section, enter your conflict of interest statement in the “Confidential to Editor” section, and submit your "Accept" recommendation.

Reviewer #1: All comments have been addressed

Reviewer #2: All comments have been addressed

2. Is the manuscript technically sound, and do the data support the conclusions?

Reviewer #1: Yes

Reviewer #2: Yes

3. Has the statistical analysis been performed appropriately and rigorously? 

Reviewer #1: Yes

Reviewer #2: I Don't Know

4. Have the authors made all data underlying the findings in their manuscript fully available?

Reviewer #1: (No Response)

Reviewer #2: Yes

5. Is the manuscript presented in an intelligible fashion and written in standard English?

Reviewer #1: Yes

Reviewer #2: Yes

6. Review Comments to the Author

Reviewer #1: (No Response)

Reviewer #2: Dear Dr. Pinheiro,

I am sorry about your loss.

Your article is now sufficiently clear. I do appreciate your work.

For the future, it would be interesting to analyze the association between comorbidity and fragility through mono- o multi-dimentional scales (like CIRS or MPI etc) and in-hospital mortality after a hip fracture, in this category of patients. Moreover, the adoption of the multidimensional scales of valuation during the hospitalization permit to get information even if the local hospital information systems have limited access to secondary diagnoses. It could be also important to evaluate if the patients undergo surgical treatment within 48 hours; this can be an important confounding factor.

Best Regards

7. PLOS authors have the option to publish the peer review history of their article (what does this mean?). If published, this will include your full peer review and any attached files.

Reviewer #1: No

Reviewer #2: No

---

## [Editor Report · Acceptance letter]

30 Sep 2020

PONE-D-19-23508R2 

Is the number of previous hospitalizations associated with increased in-hospital mortality after hip fracture in a developing country? 

Dear Dr. Pinheiro:

I'm pleased to inform you that your manuscript has been deemed suitable for publication in PLOS ONE. Congratulations! Your manuscript is now with our production department. 

Kind regards, 

on behalf of

Dr. Osama Farouk 

Academic Editor

PLOS ONE